# Subwavelength terahertz imaging via virtual superlensing in the radiating near field

Alessandro Tuniz [1,2] & Boris T. Kuhlmey [1,2]

Imaging with resolutions much below the wavelength $\lambda$ – now common in the visible spectrum – remains challenging at lower frequencies, where exponentially decaying evanescent waves are generally measured using a tip or antenna close to an object. Such approaches are often problematic because probes can perturb the near-field itself. Here we show that information encoded in evanescent waves can be probed further than previously thought, by reconstructing truthful images of the near-field through selective amplification of evanescent waves, akin to a virtual superlens that images the near field without perturbing it. We quantify trade-offs between noise and measurement distance, experimentally demonstrating reconstruction of complex images with subwavelength features down to a resolution of $\lambda/7$ and amplitude signal-to-noise ratios < 25dB between 0.18–1.5 THz. Our procedure can be implemented with any near-field probe, greatly relaxes experimental requirements for subwavelength imaging at sub-optical frequencies and opens the door to non-invasive near-field scanning.

The diffraction limit prohibits resolving features smaller than half a wavelength, as a consequence of the evanescent decay of high spatial frequencies in standard materials[1]. Conventional imaging techniques that collect light far from an object are typically bound by this limit, and much effort has been invested in developing ways to overcome it. Many techniques now provide resolutions well below the diffraction limit[2,3], relying either on near-field probing through a scanning tip[4,5], stochastic sets of scatterers or fluorophores in the immediate vicinity of the object to be imaged[6,7], or nonlinear effects allowing sub-diffraction imaging in the far field[8,9], many of which have yet to be demonstrated outside the optical spectrum. Methods to reconstruct sub-diffraction details from linear far fields also exist, but typically require some prior knowledge or assumptions on the nature of the object[10].

Lower frequency sub-wavelength imaging techniques (e.g., GHz, THz) typically rely on scanning antennas in an object's near field[11–13]. Imaging at terahertz frequencies (0.1–10 THz) would particularly benefit from any improvement in the ability to image below the diffraction limit[14], due to its many applications in biomedicine[15–18], which is hindered by established experimental challenges[19]. We refer the

reader to refs. 14,20 for a review on recent developments in terahertz imaging techniques.

There are two main techniques for sub-wavelength THz imaging: scattering-type scanning near-field optical microscopy (s-SNOM) and near-field photoconductive antennas (NFPA), with very different strengths and drawbacks: THz s-SNOM offer the state-of-the art in terms of resolution ( < 100 nm)[21,22], using a sophisticated technique in which an atomic force microscope tip is used to scatter THz radiation. It is ideal for probing local THz material properties, but can only access small-area planar surfaces and has limited signal-to-noise ratio. Information on polarization can in principle be retrieved, but requires careful interpretation[23]. Recent developments in the combined use of non-linearity and local optical excitation near s-SNOM tips could provide even further improvement in resolution[24,25]. THz s-SNOM approaches are powerful for probing THz properties locally, but are less suited to probe full 3D polarization-resolved fields, especially over larger areas such as that of dielectric or photonic crystal resonators due to the limited spatial range of atomic force microscopes.

Tips of s-SNOMs oscillate within nanometers of the object, a distance well below $\lambda/2\pi$, that is in the reactive near-field[26], where

[1]Institute of Photonics and Optical Science, School of Physics, University of Sydney, Camperdown, NSW 2006, Australia. [2]The University of Sydney Nano Institute, The University of Sydney, Camperdown, NSW 2006, Australia. ✉ e-mail: alessandro.tuniz@sydney.edu.au

evanescent waves corresponding to high spatial frequencies haven't decayed much yet[26]. This is ideal to achieve high spatial resolution, but the proximity of the tips can also affect the fields to be measured. In simple cases the tip's polarizability is small enough compared to that of the object to be imaged, so that perturbations are of first order and can readily be compensated for[23]. However for some sample geometries, the tip can cause stronger perturbations (from multiple scattering between tip and object) which require corrections beyond first order[27], and cannot simply be corrected by a deconvolution[28]. Indeed some s-SNOM techniques rely on generating a THz hot spot[29], which is an extreme perturbation of the local field distribution. Such hot-spots are beneficial to probe local THz material properties, but make measuring unperturbed fields of waveguides or resonators difficult.

In contrast, near-field photoconductive detector antennas (NFPAs)[13,30] are a practical alternative to s-SNOMS that are more suitable for probing fields than for probing nm-scale local material properties. Such NFPAs can be incorporated into any commercial THz time domain spectroscopy setup[31], and can directly probe both amplitude and phase over centimeter-squared areas with resolutions of order $10\,\mu m$ at the site of the antenna. Through the orientation of the dipole antenna, polarization can be resolved directly[20], making the method well suited to probe the full three-dimensional field of existing structured materials, waveguides or resonators. However, under standard laboratory conditions, it is common for NFPAs to be hundreds of micrometers away from the object to be imaged[32,33], due to practical alignment difficulties combined with the fact such antennas are expensive and delicate, making near-contact scans a risky proposition. At THz frequencies, distances that are hundreds of micrometers from an object are associated with its radiating near-field region, corresponding to distances between $\lambda/2\pi$ and $\lambda$[26] for sub-wavelength objects. The radiating near-field is characterized by fields that are dominated by radiating rather than evanescent waves. In this region, the fields don't decay as $1/r$ yet as they do once in the far field, while high spatial frequencies decay significantly, preventing genuine sub-wavelength imaging[1]. Such increased measurement distances, while leading to a loss in resolution, have the benefit of reducing perturbations to the fields due to the probe itself, which can be important given NFPAs and their substrate are considerably larger than s-SNOM tips.

Be it with s-SNOMs or NFPAs, if one wants to resolve the near-field of existing structures, ideally one would image far away enough to avoid probe- object interactions, but close enough to avoid decay of evanescent waves. This seeming contradiction could be achieved by imaging in the radiating near-field if one had a way to reverse, or avoid, the exponential decay of the evanescent field. This can be achieved with so-called superlenses (SL)[34,35] and hyperlenses (HL)[36], which respectively amplify or propagate the evanescent fields, and which have been demonstrated over much of the electromagnetic spectrum[11,35,37–41]. An ideal superlens with unlimited evanescent gain would allow imaging at any distance without loss of resolution, and thus enable non-invasive imaging of the near-field. In contrast, a hyperlens needs to be placed in the reactive near-field to resolve high spatial frequencies, and thus a priori does not help with non-invasive near-field imaging. Both approaches still carry two challenges: (i) the resolution of the highest spatial frequencies is adversely affected by even modest material losses[42]; (ii) most geometries transfer the near-field information without spatial magnification which converts evanescent waves into propagating waves[38,43]: fields must still be measured in the near field of the SL or HL[44], shifting the problem rather than solving it. At THz frequencies, anisotropic metamaterials have been used for sub-wavelength propagation of near-field information across finite slabs[38,45,46] using Fabry-Perot resonance-induced evanescent amplification[47,48], but a THz SL which truly amplifies decayed evanescent fields has so far been beyond reach.

In principle, in the absence of noise, one could imagine reversing the evanescent decay of high spatial frequencies without physical devices, using numerical amplification and phase reversal instead: evanescent waves contribute to the field measured at any distance, superimposing high spatial frequency fluctuations on top of low spatial frequency radiating fields. If these minute amplitude and phase fluctuations can be resolved, they can also be amplified to regain the original field. To our knowledge, such a scheme has not been considered, presumably because high spatial frequencies rapidly decay well below instrument noise.

Here we show that practical noise levels allow for a useful extraction of decayed information in the radiating near-field region, also providing a way for increasing the resolution in the reactive near-field region. In effect, we experimentally demonstrate a virtual superlens through post-processing, reconstructing previously indiscernible sub-wavelength spatial features contained within complex images, with demonstrated resolution down to $\lambda/7$ taken from a distance greatly reducing field perturbation by the probe. Our approach is general, provided that low-noise, phase-resolved fields can be measured. This demonstrates the possibility of measuring near fields without perturbing them – which would be particularly useful when imaging fields in structures that are sensitive to perturbations, e.g., high-Q/topological resonances[49,50], photonic crystal defects[51] and nanoresonators[52].

## Results

### Approach and implementation

Figure 1 shows a schematic of our approach, which aims to image a planar source object possessing sub-wavelength features with a field $\mathbf{E}^{obj}(x, y, z = 0)$. The total field at $\mathbf{r} = (x, y, z)$ is given by a Fourier expansion[34]

$$\mathbf{E}(\mathbf{r}) = \sum_\sigma \iint_{k_x, k_y} \tilde{\mathbf{E}}^\sigma(k_x, k_y) \exp(i\mathbf{k} \cdot \mathbf{r}) dk_x dk_y, \qquad (1)$$

where $\sigma$ sums over polarizations, $\mathbf{k} = (k_x, k_y, k_z)$,

$$k_z = (k_0^2 - k_x^2 - k_y^2)^{1/2}, \qquad (2)$$

$\tilde{\mathbf{E}}^\sigma$ can be obtained from the Fourier transform of $\mathbf{E}^{obj}(x, y, z = 0)$, and $k_0 = 2\pi/\lambda$ is the free space wavenumber. Propagating waves (Fig. 1,

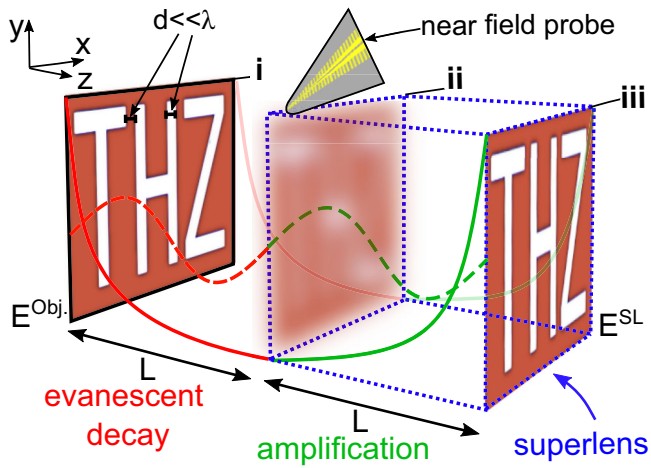

**Fig. 1 | Concept schematic of virtual superlens. i** Sub-wavelength spatial features are carried by evanescent waves which exponentially decay over a $L$ (red). **ii** The resulting lower-resolution image is detected by a near-field probe. The collected evanescent fields are then numerically amplified over $L$ (green), leading to **iii** the original image, analogously to a superlens (blue). Wavelength-scale information is carried by propagating waves (dashed).

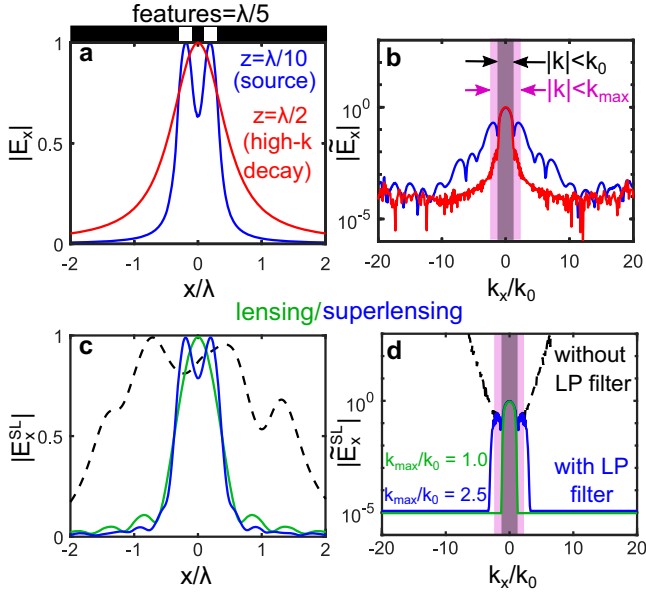

**Fig. 2 | Numerical example of virtual SL. a** An $x$-polarized field is incident on sub-wavelength apertures ($d = \lambda/5$, blue), which cannot be discerned at $z = \lambda/2$ (red). **b** Associated spatial Fourier transform. Black/purple regions are propagating/evanescent. **c** Images after virtual lens ($k_{max} = k_0$, green), after SL using the full spectrum (black), and using the low-passed (LP) filtered spectrum ($k_{max} = 2.5k_0$, blue). **d** Associated spatial Fourier transforms.

dashed lines) carry information emerging from spatial frequencies satisfying $k_x^2 + k_y^2 < k_0^2$ and impose a lower limit on the spatial features $d$ which can be resolved in the far field. Evanescent waves (Fig. 1, solid lines) carry sub-wavelength spatial frequencies satisfying $k_x^2 + k_y^2 > k_0^2$ and exponentially decay in free space. As a result, the fine details of an image possessing spatial features $d \ll \lambda$, detected by a near-field probe at a distance $z = L$, cannot be resolved. This process can be straightforwardly reversed via the transformation $(x, y, z) \to (-x, -y, -z)$ over a subsequent length $L$, by numerically reversing the phase accumulated by the propagating waves and amplifying the evanescent waves (green curves in Fig. 1). In practice, we measure an $x$-polarized field $E_x(x, y, z = L)$. Starting from Fourier components of the measured field $\tilde{E}_x(k_x, k_y)$, the electric field after the virtual SL is given by:

$$E_x^{SL}(x, y) = \iint_{k_x, k_y} \tilde{E}_x(k_x, k_y) \exp\left[-i(k_x x + k_y y + k_z L)\right] dk_x dk_y, \quad (3)$$

where $k_z \in \mathbb{C}$ follows Eq. (2), with arbitrarily high spatial frequencies. For large spatial frequencies, $k_z$ is imaginary and leads to exponential amplification $\exp|k_z|L$. This process is equivalent to "superlensing"[34], achieving the same effect (Fig. 1, blue dotted line). Simply reversing the phase without amplifying evanescent waves, that is limiting the integrals to real $k_z$, is akin to what an ideal conventional far field lens achieves (here termed "lensing").

In Eq. (3) higher spatial frequencies lead to larger amplification terms so that, after amplification, high spatial frequency noise is bound to dominate over any signal at lower spatial frequencies. A simple numerical example showcases the issue: we consider 2D finite element method calculations where the domain is infinite in $y$, with TM polarized fields (non-zero magnetic field in $y$). Figure 2a shows $|E_x|$ emerging from two perfectly conducting apertures with width- and edge-to-edge- separation of $d = \lambda/5$. At a distance $z = \lambda/20$ in the reactive near-field region, the two apertures can be distinguishe (blue curve in Fig. 2a). At a distance of $z = \lambda/2$ in the radiative near field region (red) this is no longer the case. Figure 2b shows the associated spatial Fourier transforms. The minimum magnitude of $k_x$ required in order to

resolve $d$ is given by $k_x/k_0 = \lambda/2d = 2.5$. While the source's spatial Fourier spectrum extends beyond $k_x = 10k_0$, at $z = \lambda/2$ (i.e., in the radiating near-field region) most field components with spatial frequencies inside $|k| < k_{max}$ have exponentially decayed 20 dB below that of propagating waves, so that the apertures cannot be distinguished. Note that while the source spectrum (Fig. 2b, blue) is smooth, the radiating near-field spectrum (Fig. 2b, red) presents numerical noise starting even at modest $k_x/k_0$ values. Using this noise level compared to the maximum signal, the amplitude signal-to-noise (SNR) is -30 dB.

We now implement the superlens procedure given by Eq. (3) to the complex field associated with the red curves in Fig. 2a, b obtained at $z = \lambda/2$, thus taking $L = \lambda/2$. The result is shown as a black dotted line in Fig. 2c. The image reconstruction of the two apertures has clearly failed: the associated spatial Fourier spectrum (black line in Fig. 2d) shows that noise at high spatial frequencies has been amplified to exceed the amplitude of any signal, including that of the propagating waves. However, not all is lost: comparing the black curve amplified spectrum with the blue curve in Fig. 2b, the amplified spectra match with the original for $k_x/k_0 \lesssim 2.5$, where the signal in the evanescent spectrum was originally above the noise level (Fig. 2b, purple). We thus apply a spatial low-pass filter in $k$-space after superlensing, setting all $|k_x| > 2.5k_0$ to zero. The result is shown as blue curves in Fig. 2c, d, where we find that the virtual SL now resolves the apertures as desired. In contrast, filtering all non-propagating waves as per a conventional lens, i.e., setting regions where $k_x > k_0$ to zero (green curves in Fig. 2c, d), does not allow us to resolve the apertures, even though the phase is reversed in the procedure.

This example shows there is much to be gained from amplifying evanescent waves, provided amplification is limited to spatial frequencies with signal above the noise floor. In the above example, the spatial frequency limit imposed by the numerical noise was $2.5k_0$, but this clearly depends on the actual signal-to-noise ratio: By equating the signal-to-noise ratio with the amplification factor $\exp(k_z z)$ for a signal measured at a distance $z$, it can be shown (see Supplementary Note 1 for the derivation) that the maximum useful spatial frequency is

$$k_{max} = k_0 \sqrt{1 + \left(\frac{\lambda}{z} \frac{\log 10}{20\pi} SNR\right)^2}, \quad (4)$$

where SNR is the experiment's signal-to-noise ratio (in dB) of the amplitude $E_x$. Filtering out spatial frequencies above this limit reduces the spatial resolution, but avoids exponential increase of noise, leading to improved images. Equation (4) directly estimates the maximum spatial frequency that can be retrieved, and thus the resolution that can be achieved, at any $z$ and SNR. Increasing $z$ could thus be advantageous – reducing the near-field perturbation induced by the antenna – provided SNR increases accordingly.

## Noise/resolution trade-off

We now consider an example to showcase the implications of Eq. (4) by expanding our analysis on the simulations associated with Fig. 2. Figure 3a (left) shows a simulation of the amplitude $E_x(x)$ as a function of normalized propagation length $z/\lambda$ for the double aperture case of Fig. 2, before any superlens procedure. Note in particular that inside the interval $(2\pi)^{-1} < z/\lambda < 1$, i.e., in the radiating near field, the two apertures are not resolved. On the right of Fig. 3a we show the target amplitude at the source, that is, what the apertures should look like after perfect, noiseless superlensing: both apertures are resolved for all values of $z/\lambda$. The blue line in Fig. 3b shows the Fourier transform $|\tilde{E}_x|$ at a distance $z/L = 0.5$, as per Fig. 2b. To enable a clear analysis, we add random amplitude and phase, resulting in a nominally flat SNR of 30 dB, shown as a line curve in Fig. 3b. We then perform the superlensing procedure without any filtering, for different propagation lengths $z$, always adding white noise such that SNR = 30 dB before amplification. The resulting $|\tilde{E}_x^{SL}(k_x, k_y)|$, normalized to $|\tilde{E}_x^{SL}(0,0)|$, is

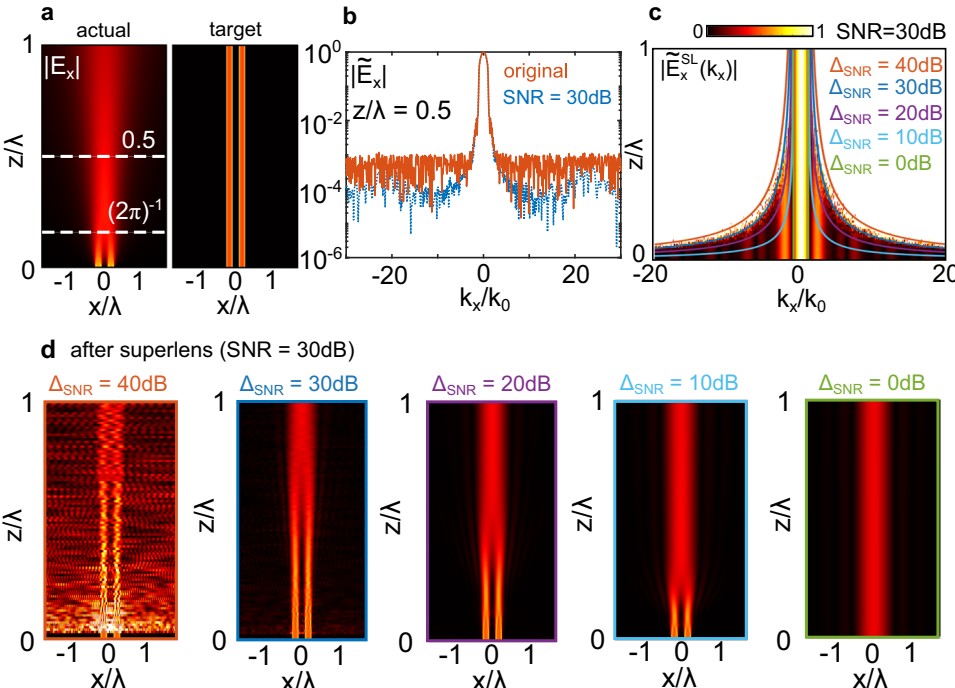

**Fig. 3 | Numerical example illustrating the effect of the SNR. a** Left: raw simulation of the amplitude $E_x(x)$ as a function of $z/\lambda$ for the double aperture case of Fig. 2, before any superlens procedure. Right shows the target amplitude at the source. **b** Dashed blue line shows an example $|\tilde{E}_x|$ at a distance $z/\lambda = 0.5$, as per Fig. 2b. The red line shows the same field after adding random amplitude and phase resulting in a flat SNR of 30 dB. **c** Calculated normalized amplified spatial Fourier transform for the flat SNR = 30 dB, as a function of its normalized propagation length $z/\lambda$. Color scale has been saturated to 1 for clarity. Solid

curves show Eq. (5) choosing different values of $\Delta_{SNR}$ as labeled. Note that the case $\Delta_{SNR} = SNR = 30$ dB corresponds to the boundary between where high spatial frequencies have a comparable magnitude to propagating waves. **d** Resulting $E_x^{SL}(x)$ after applying the superlens procedure, using a low-pass filter function bounded by $k_x/k_0$ as per Eq. (5) for different values of $\Delta_{SNR}$ as labeled. If $\Delta_{SNR} > SNR$, the aperture images are plagued by noise; if $\Delta_{SNR} \leq SNR$, the maximum distance at which the retrieval procedure produces the image is gradually reduced.

shown in Fig. 3c, where the color scale is saturated at unity. For short lengths $z/\lambda$, the procedure yields the required high spatial frequency components. For increasingly long propagation lengths however, the amplitude of amplified noise of evanescent waves becomes greater than those of the propagating waves (white, saturated regions). Equation (4) predicts the boundary between these two regions for SNR = 30 dB. To show this, Fig. 3c also shows contour lines of the function

$$k_x/k_0 = \sqrt{1 + \left( \frac{\lambda}{z} \frac{\log 10}{20\pi} \Delta_{SNR} \right)^2}, \qquad (5)$$

for different choices of $\Delta_{SNR}$ as labeled. For $\Delta_{SNR} = SNR = 30$ dB (dark blue), where the $\Delta_{SNR}$ value used in Eq. (5) is that of the actual SNR, Eq. (5) indeed yields the spatial frequency boundary inside which the evanescent field amplitude after the virtual SL remain below that of propagating waves. Decreasing $\Delta_{SNR}$ narrows the available range of $k_{max}$, as expected, with $\Delta_{SNR} = 0$ dB corresponding to $k_{max} = k_0$. Given a set of experimental conditions, Eq. (4) can thus be used as a first estimate of the low-pass spatial frequency boundary to include after the superlensing procedure. The cutoff can then be fine-tuned until a suitable image is obtained.

Virtual superlensing thus involves a trade-off between image sharpness and noise. The better the signal-to-noise ratio of the measurement, the higher a resolution is possible. For a given signal-to-noise ratio, using a higher spatial frequency cutoff than predicted by Eq. (4) has the effect of deteriorating the image by introducing amplified noisy spatial frequencies, as shown in Fig. 3d, left. In contrast, decreasing the spatial frequency filter cutoff below Eq. (4)

removes high spatial frequencies needed to resolve fine feature sizes, as shown in Fig. 3d, right.

Equation (4) can also be used to estimate the largest distance $z$ at which a certain resolution can be achieved: this can be useful when non-invasive imaging is desired. For example in Fig. 3, the minimum spatial frequency required to distinguish the apertures is $k_{max}/k_0 = 2.5$. Rearranging Eq. (4) for SNR = 30 dB indicates the required resolution can be imaged from up to $z/\lambda \simeq 0.48$, which matches the largest $z/\lambda$ for which the double aperture is distinguished in Fig. 3d for $\Delta_{SNR} = 30$ dB.

## Experiments

Figures 4 and 5 showcase our technique on two distinct imaging experiments. Our experiment uses a commercial pulsed THz source (Menlo TERAK15, 0.1–3 THz). Lenses collimate and focus the THz beam towards patterned laser-machined samples containing sub-wavelength features ($d = 150 - 200\,\mu m$). The transmitted field amplitude is sampled as a function of the time delay of a probe pulse on a photo-conductive antenna (Protemics TD-1550-X-HR-WT-XR) which probes the radiating near field. The electric field is polarized in $x$, using the reference frame shown in Fig. 1. Diffraction limited imaging of the finest feature would require a wavelength of $\lambda/2 = d$, i.e., frequencies of 0.75 – 1 THz. See Supplementary Fig. 1 for detailed images of samples and near-field probe.

Figure 4a, b shows the measured intensity $|E_x|^2$ emerging from two apertures (diameter and separation: 200 $\mu m$) at a frequency of 1.5 THz and 0.38 THz respectively, as well as the associated spatial Fourier transform magnitude $|\tilde{E}_x|$. Figure 4c shows the average measured intensity across $y = 0 \pm 100\,\mu m$ as a function of $x$ and frequency, and highlights that apertures cannot be discerned directly: at higher frequencies, what would be measured by a scanning antenna or tip is a

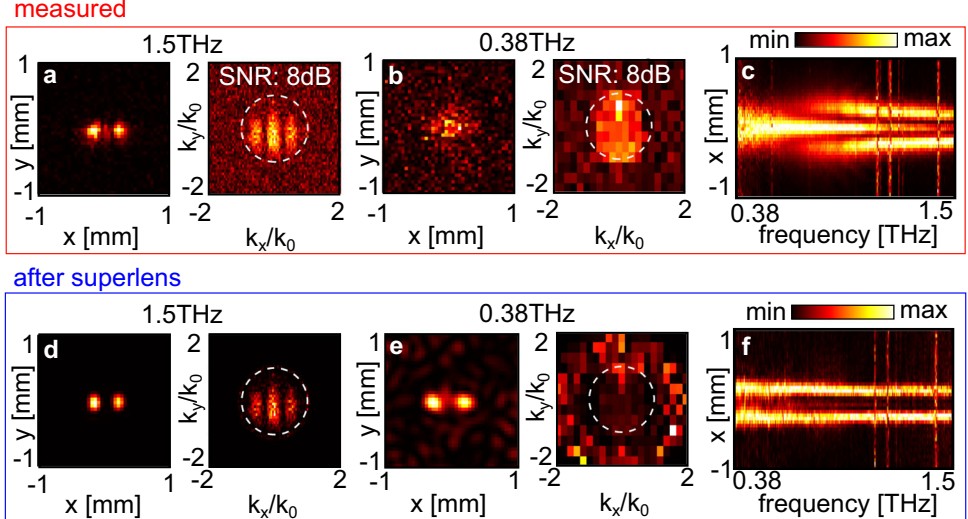

**Fig. 4 | Superlens experiment, imaging two apertures.** Measured $|E_x|^2$ and $|\tilde{E}_x|$ for two apertures of diameter/separation $d = 200\,\mu m$, at **a** 1.5 THz and **b** 0.38 THz, with SNR as labeled. Dashed white circles show $|k| = k_0$. **c** Corresponding intensity profile in $x$ as a function of frequency averaged over $y = 0 \pm 100\,\mu m$. **d** $|E_x^{SL}|^2$ after the superlens at 1.5 THz and **e** 0.38 THz. **f** Corresponding intensity profile in $x$ as a function of frequency averaged over $y = 0 \pm 100\,\mu m$. Vertical artefacts in **c**, **f** are absorption lines due to air humidity.

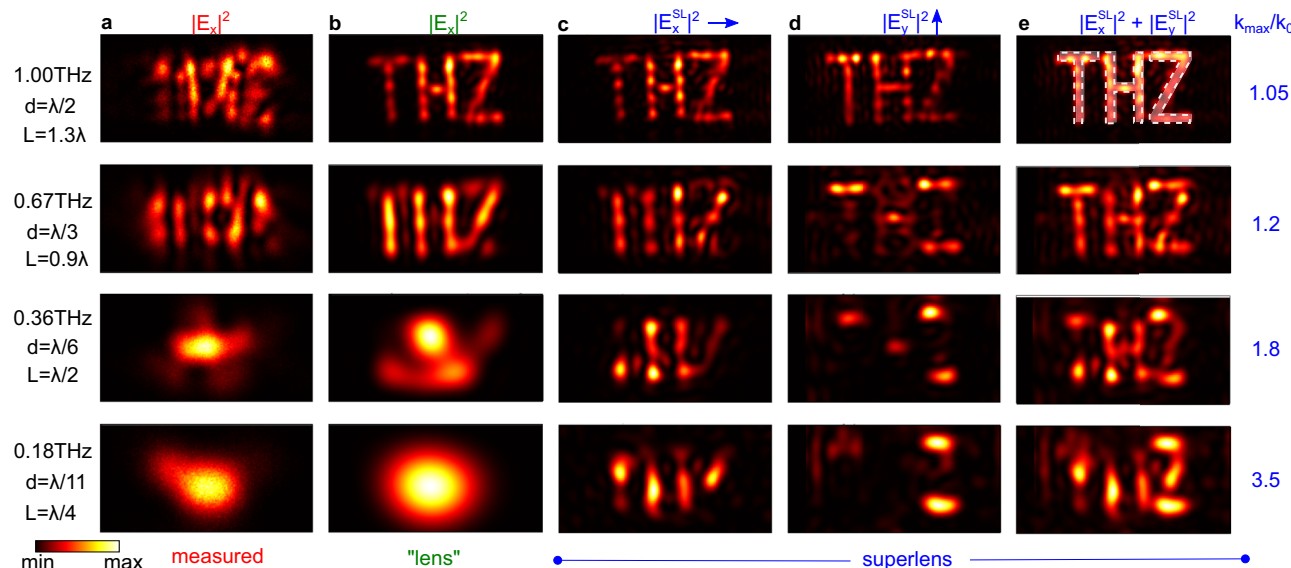

**Fig. 5 | Superlens experiment, imaging the letters "THZ". a** Measured $|E_x|^2$ at different frequencies labeled left, with associated $d$ and $L$ in terms of $\lambda$. **b** Corresponding $|E_x|^2$ after image reconstruction with $k_{max} = k_0$ (lens) and **c** when $k_{max} > k_0$ ($|E_x^{SL}|^2$, superlens). **d** $|E_y^{SL}|^2$ after the superlens with $k_{max} > k_0$. **e** Reconstructed images using measured $x-$ and $y-$polarized fields in (**c**, **d**). The ratio $k_{max}/k_0$ for each row is shown on the right. Each window area is 4 mm × 2 mm. The smallest feature size is $d = 150\,\mu m$ and the detector distance is $L = 440\,\mu m$.

diffraction pattern that includes additional features, requiring phase reversal (i.e., lensing) to reconstruct an accurate image. At lower frequencies the evanescent decay blurs out the features of the double aperture.

We calibrate the probe-to-sample distance $L$ by considering the complex field at a frequency above the diffraction limit (here: 1.5 THz), and adjusting $L$ in Eq. (3) to maximize image sharpness (see Supplementary Fig. 2). From $|\tilde{E}_x|$, we then obtain the frequency-dependent SNR via the ratio between the maximum amplitude in the propagating region $|\mathbf{k}| < k_0$, and the average amplitude in the evanescent region $|\mathbf{k}| > k_0$, from which we estimate the experimentally accessible $k_{max}$ via Eq. (4). For the two-aperture experiment we find $L = 172\,\mu m$ (i.e. $L \simeq 0.87\lambda$ at 1.5 THz and $L \simeq 0.22\lambda$ at 0.38 THz) and SNR = 4–14 dB between 0.38–1.5 THz, resulting in $k_{max}/k_0 = 1 - 1.9$

(see Supplementary Fig. 3). We then implement the SL via Eq. (3), followed by spatial low-pass filtering bounded by $k_{max}$. Figure 4d, e show $|E_x^{SL}|^2$ and $|\tilde{E}_x|$ at 1.5 THz and 0.38 THz respectively: the apertures are now resolved. The associated average intensity over the middle of the two apertures (Fig. 4f), shows this procedure works over the entire THz frequency band.

Finally, we implement our procedure on a complex, large-area image formed by a laser-written metal sheet containing the letters "THZ" (minimum feature size: $d = 150\,\mu m$). In this case, the calibration yields $L = 440\,\mu m$, SNR = 15–25 dB, and $k_{max}/k_0 = 1 - 3.5$ between 0.2–1 THz (see Supplementary Fig. 3). Figure 5a shows the measured $|E_x|^2$ at different frequencies as labeled. Figure 5b shows the corresponding retrieved field $|E_x^{SL}|^2$ through Eq. (3) at different frequencies with $k_{max} = k_0$, i.e., a conventional lens simply reversing phase. At

1.0 THz (i.e., the diffraction limit), the letters "THZ" can be discerned; at 0.67 THz, only the vertical features are resolved; lower frequencies do not provide a sufficiently sharp image to discern the finer features of the original image, with only a single large spot occurring at 0.18 THz. Figure 5c shows the corresponding retrieved $|E_x^{SL}|^2$ at different frequencies with $k_{max}/k_0$ as labeled: vertical features are significantly sharpened. Note horizontal features do not let $E_x$ through, because the slits forming the letters act as parallel plate waveguides with width smaller than half a wavelength, in which solely the TEM mode polarized perpendicularly to the thinnest features can propagate. Therefore, thin features in $y$ ($x$) only appear for the $E_x$ ($E_y$) field. As a result, for $x$ − polarized fields the letters' vertical ($y$-oriented) features are clearest, and clearly show different numbers of nodes and antinodes for different frequencies, revealing the frequency-dependent modal field structure inside each letter in excellent agreement with simulations (see Supplementary Fig. 4). We repeat the above procedure for a polarization oriented in $y$ relative to the sample orientation, and plot the corresponding $|E_y^{SL}|^2$ in Fig. 5d to resolve the horizontal features of each letter.

The $E_x$ and $E_y$ components in Fig. 5c, d have the richest information on the actual, unperturbed local electric fields at the aperture site, but if the shape of the aperture rather than the fields is desired, a full image of the aperture can be obtained by summing the two contributions. Figure 5e shows the resulting $|E_x^{SL}|^2 + |E_y^{SL}|^2$ distribution, clearly showing the emergence of the letters "THZ" at all frequencies, down to $\lambda/7$. These results are in agreement with simulations of the transmitted sub-wavelength pattern (see Supplementary Fig. 4). Remarkably, the resolution achieved in Fig. 5 is higher than that in Fig. 4, even though the near field antenna's distance to the object was more than doubled, thanks to a higher SNR exceeding the loss from increased evanescent decay.

## Discussion

In this paper, we have presented a superlensing approach which numerically amplifies measured evanescent fields. Spatial resolution is then limited by a trade-off between measurement distance and signal-to-noise ratio, rather than by distance alone. We presented experiments illustrating the process at THz frequencies using commercially available facilities. Compared to previous superlens incarnations[34,35,42], our approach circumvents losses altogether by removing the need for materials: the evanescent fields are measured in air rather than after a structured material, and the reversal of decay is achieved numerically instead. While our approach is particularly well suited to THz near-field photoconductive setups, it can be adapted to suit any near-field experiment which measures amplitude and phase, immediately providing a pathway for increasing the imaging resolution of near-field setups at any frequency, or reduce probe perturbation while maintaining resolution. Indeed, since our technique enables near-field measurements in the radiating rather than reactive near field, it allows for accurate near-field imaging without perturbing the intrinsic field of structures. This will be particularly useful for imaging fields strongly susceptible to local disturbances, such as those of high-Q/topological resonators[49,50] and photonic crystal defects[51], see Supplementary Fig. 7 for simulations showing the effect of perturbation on a high-Q silicon resonator. A simple perturbative analysis, presented in Supplementary Note 2, shows that the perturbation of resonant frequency can be lowered by orders of magnitude without loss of spatial resolution. This could be of use at frequencies outside the terahertz range: in a recent paper, Esmann et al.[52] showed how their exquisite optical SNOM tip could be used to resolve near-infrared fields of a 40 nm gold nano-resonator with 5 nm resolution – however, contrast could only be achieved at tip distances below 10 nm, leading to strong coupling and perturbation of the resonances themselves. Using the virtual lens processing of the same data, surface fields of the unperturbed resonance

could potentially also have been retrieved. Measuring the near-field at resonance without probe-induced perturbations could help to uncover scattering and loss mechanisms which limit higher quality resonances, while offering avenues to explore more complex near-field interactions such as Förster resonance energy transfer (FRET)[53].

## Methods

### Experimental setup

A summary of samples and photographs of the experimental setup is shown in Supplementary Fig. 1. We use a commercially available THz-TDS System (Menlo TERAK15), which relies on THz emission from biased photoconductive antennas that are pumped by fiber-coupled near-infrared pulses (red line; pulse width: 90 fs; wavelength: 1560 nm). Terahertz lenses collimate and focus the beam towards the sample. The THz field emerging from the aperture is sampled as a function of the time delay of a fiber-coupled probe pulse on another photoconductive antenna (Protemics TD-1550-X-HR-WT-XR), which forms the THz detector. The electric field is polarized in $x$, using the sample orientation and reference frame shown in Fig. 1. A moveable, fiber-coupled near-field detector module enables the measurement of the $x$-polarized electric field at the output of the laser-machined samples. The near-field is spatio-temporally resolved at every point via a raster scan (step size: 25–50 $\mu$m). Fast Fourier transforms of the temporal response at each pixel position provide the spectral information, including amplitude and phase (spectral resolution: 5–8 GHz). Further details of the experimental setup are described in ref. 33. The phase fluctuation, obtained from the standard deviation of the phase of the Fourier transform of 850 consecutive measurements, is between 0.05–0.35 rad between 0.2–1.5 THz, as shown in Supplementary Fig. 5a–c, which has no discernible effect on the superlens imaging quality for our experimental conditions, as shown in Supplementary Fig. 5d–f.

## Data availability

The data used to produce the plots within this paper is available at https://doi.org/10.6084/m9.figshare.23633757. Any other data and findings of this study are available from the corresponding author on request.

## Code availability

The code used to produce the plots within this paper is available at https://doi.org/10.6084/m9.figshare.23633757.

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

## Acknowledgements

This work is funded in part by the Australian Research Council Discovery Early Career Researcher Award DE200101041 (A.T.). The authors thank Angus Michael O'Grady, Gleb Kozlov and Giuseppe Della Valle for fruitful discussions. The authors thank Benjamin Johnston from the Optofab Node of the Australian National Fabrication Facility for fabricating the laser-machined samples.

## Author contributions

A.T. and B.T.K. conceived the idea. A.T. performed the experiments and simulations. B.T.K. derived the noise dependent maximum spatial frequency resolution limit. A.T. and B.T.K. wrote the manuscript. A.T. directed the project.

## Competing interests

The authors declare no competing interests.
