## [Peer Review File · Nature Communications]

Subwavelength terahertz imaging via virtual superlensing in the radiating near fieldREVIEWER COMMENTS

Reviewer #1 (Remarks to the Author):

This manuscript describes an approach for implementing a computational version of a superlens in order to obtain sub-wavelength imaging without placing a near-field probe in close proximity (much closer than λ) to the surface being imaged. It is of particular interest for the long-wavelength regime, where direct measurements of the electric field (i.e., obtaining both amplitude and phase) are feasible. The authors observe that it may be valuable in certain circumstances to avoid the use of a near-field probe because it can be challenging to deconvolve the effects of an antenna or scattering object that has been inserted into the near field regime. Their method involves measuring the field (both amplitude and phase) at an intermediate distance (e.g., about one wavelength away), where the evanescent fields have decayed significantly (but not completely), and then numerically windowing out the spatial frequency components which are not needed to reconstruct the image (because they are susceptible to contamination by noise). The experiments demonstrate image reconstruction with a resolution of as good as $\lambda/7$, even though the measurement location (distance from the sample) at that wavelength is $\lambda/4$, well outside the range where evanescent fields could be directly detected.

This is an interesting technique, which builds on a considerable body of earlier work involving superlenses and hyperlenses in a fascinating new direction. That large body of literature has so far had little impact at these low frequencies, but this work could build a useful bridge between THz science and those earlier efforts. Therefore I am in general favorable towards publication of this work.

My primary concern is that the demonstrated value of the technique is not sufficiently clear in the manuscript. If we just look at the raw numbers (spatial resolution of $\lambda/7$ without having to approach the sample closer than $\lambda/4$), it just doesn't seem that the technique offers a huge benefit when compared to more well-known probe-based techniques (e.g., typical s-SNOM measurements achieve $\lambda/10,000$ by approaching the sample within 10 nanometers). I agree that the aforementioned issue of deconvolution of the effect of the near-field probe is sometimes an issue; however, I must also point out that this issue has been the topic of many publications in recent years, and to a great extent is now considered a "solved" problem. (I write "solved" in quotation marks because it isn't rigorously solved for all situations; but, everyone thinks that it is at least manageable for most situations.) Therefore, one is led to ask: why should one make a big sacrifice in spatial resolution just to avoid having to deal with a problem that isn't really much of a problem in many cases? One answer, of course, lies in those few cases where deconvolving the effect of the near-field probe REMAINS a problem. This manuscript would be much more powerful if the authors devoted some time justifying the need for their approach, perhaps by focusing on these sorts of situation. Or, alternatively, some time speculating about how much farther their technique could be pushed (in terms of the trade-off posed here: spatial resolution vs. the distance between the sample and the measurement point).

A few minor points to also consider:

1) I am confused about the following sentence:

"Figure 3(c) shows the average measured intensity across $y = 0 \pm 100 \mu\text{m}$ as a function of x and

frequency, and highlights that apertures cannot be discerned directly, either due to diffraction at higher frequencies, or because of evanescent decay at lower frequencies."

To me, it looks as if the two apertures are indeed well resolved at higher frequencies (although obviously not at lower frequencies). For example, it seems pretty clear when looking at Fig. 3(c) at 1.5 THz that there are two apertures, despite the weaker diffraction bump in between them. Not as clearly as in Fig. 3(f), but even so. The authors should clarify what they mean here.

2) On page 4, top paragraph, second line from the end, where it says this: "i.e., setting regions where $k > k_0$ to zero" Is that supposed to be " $k_x > k_0$ "? Either that's a typo (with the subscript x accidentally omitted on the left hand side of the inequality) or I am confused by that sentence.

3) I would mildly take issue with this phrase that appears in the very first paragraph of the document: "or nonlinear effects [8, 9], only really possible in the optical spectrum." Several groups have demonstrated nonlinear effects in near-field imaging and spectroscopy at terahertz frequencies. Indeed, there have even been suggestions that such nonlinearities can lead to improved spatial resolution compared to linear measurements. See, e.g., this reference: <https://doi.org/10.1364/OE.382130>

Reviewer #2 (Remarks to the Author):

Authors of this paper realised a simple but efficient and original approach for recovering some of the information about the sub-wavelength object caused by the decay of the evanescent waves. The topic of super-lenses was active over 20 years ago, and it is based on the principle of recovering the evanescent waves that form a near field of the object and thus cannot be imaged in the far field of the source. Those super-lenses used physical mechanisms to enhance the amplitude of evanescent waves. In present work, authors proposed to numerically enhance the signals carried by the evanescent waves, provided their amplitude is still above the noise level. They have shown a visible enhancement of the image quality for when the fields are measured in a sort of an intermediate zone of radiation. I believe that this is a very useful approach, and can be readily used even in near field scanning systems for enhancing the resolution (that is also limited due to the finite distance from the probe to the sample).

Reviewer #3 (Remarks to the Author):

This manuscript presents a near-field scanning technology, which can tolerate a farther distance between the detection probe and the samples, with a post-processing calculation. The experiment demonstrates a resolution of $\lambda/7$ and amplitude signal-to-noise ratios below 25dB between 0.18–1.5 THz. However, the authors may realize this method at the cost of noise sensitivity and resolution limits. The comments are in the following:

1. The innovation should be further clarified:

(1) This manuscript is essentially near-field scanning imaging, which does not avoid the process of mechanical scanning. Besides, although the detection distance is farther, it still needs to directly detect evanescent waves, which belongs to near-field imaging. So, what are the advantages of this work compared with terahertz near-field scanning imaging employing the AFM probes? The SNOM with an AFM probe can realize a very high resolution without the post-process method.

(2) The proposed challenge “the resolution of the highest spatial frequencies is adversely affected by even modest losses” in the introduction seems to be shifted or unsolved in this work. The resolution in this work seems to be restricted by the noise, which is caused by the post-process method. Only low-loss conditions can be resolved.

2. Some technical details should be further clarified:

(1) To filter noise, the authors apply a low pass filter in k space. In this condition, the resolution is confined, since the high-k information is also lost. (for $k > 2.5k_0$). How to solve this problem?

(2) This method needs the phase information for the post-process. How can the phase be accurately measured in the terahertz region? How does the phase deviation affect your imaging?

(3) The authors image the “THZ” words in their experiment, is it still effective to super-resolution image other samples? The scattering spectrum by different samples will be different, so the low-pass filter may have different ranges. This issue needs to be further explained.

(4) The realization of the spatial Fourier transforms in Figure 2,3 should be explained. The statement should include the step size of the space domain ($\Delta x, \Delta y$), the step size of the k domain, and the relationship between k_{max} and $\Delta x, \Delta y$.

3. Some statements should be further explained:

(1) What is the difference between the reactive near-field region, radiating near-field region, and far-field region in terahertz imaging? Please explain their spatial division and analyze the characteristics of each area.

(2) In the abstract, the authors say “At lower frequencies, exponentially decaying evanescent waves must be measured directly, requiring a tip or antenna to be brought into very close vicinity to the object.” At terahertz frequencies, the evanescent waves can also be detected in the far field, without a tip or antenna. These conditions should not be excluded.

Reviewer comments and requests:

We thank the Reviewers for their constructive feedback, which has greatly improved the quality of the manuscript. We wish to point out that Figures 2,3,4,5 and Suppl. Fig. 5,6 can be reproduced with the MATLAB code and data which can be found at the following private and confidential link: <https://figshare.com/s/91a77322ddbfb08985ad>. A DOI has been reserved at the following link, <https://doi.org/10.6084/m9.figshare.23633757>, and this code will be made public after manuscript acceptance.

Reviewer #1

R1.0a. This is an interesting technique, which builds on a considerable body of earlier work involving superlenses and hyperlenses in a fascinating new direction. That large body of literature has so far had little impact at these low frequencies, but this work could build a useful bridge between THz science and those earlier efforts. Therefore I am in general favorable towards publication of this work.

We thank the Reviewer for the positive feedback.

R1.0b. My primary concern is that the demonstrated value of the technique is not sufficiently clear in the manuscript.

We see the reviewer's excellent point, and have significantly extended our discussion of where, and where not, our technique is particularly beneficial, both in the introduction and in the conclusion, and in the Supplementary Information – see detailed responses in the points R1.0c, R1.0d below and blue-lined version of the manuscript.

R1.0c. If we just look at the raw numbers (spatial resolution of $\lambda/7$ without having to approach the sample closer than $\lambda/4$), it just doesn't seem that the technique offers a huge benefit when compared to more well-known probe-based techniques (e.g., typical s-SNOM measurements achieve $\lambda/10,000$ by approaching the sample within 10 nanometers). Why should one make a big sacrifice in spatial resolution just to avoid having to deal with a problem that isn't really much of a problem in many cases?

At THz frequencies there are two main near-field scanning imaging setups: The first type, based on SNOM tips which scatter the near-field with a nanometer-sized oscillating tip for phase locked far-field detection, excels at achieving incredible spatial resolutions by approaching the sample within 10 nm, but suffers three notable drawbacks: (i) it requires a dedicated s-SNOM (or AFM) that is a rather pricey (and still rare) addition to any THz set up; (ii) the imaging area is typically limited to micrometers squared; and (iii) the signal-to-noise is relatively low. In some forms of s-SNOM, excitation of the THz radiation is localized at the tip through optical excitation, or through hot-spot effects of the tip. In these local excitation version, s-SNOM is particularly powerful to measure local THz properties with nm-scale resolution, but is not amenable to measuring *existing* fields, say of a waveguide or resonator mode. The second main type of near-field scanning imaging setups uses a photoconductive antenna that is itself placed in the near field. Such antennae are a cost-effective way to add near-field imaging capabilities to any existing THz-TDS system, can offer substantially better signal-to-noise than s-SNOM based imaging, can scan much larger areas (millimetres squared) and can measure the field polarization in all three directions. The drawback of photoconductive near field antennae is that they

are relatively large and difficult to bring very close to samples, severely limiting resolution, although the effect of antenna size alone can be removed through deconvolution techniques. Photoconductive near-field antenna imaging is well suited to study the near field of larger samples, such as THz photonic devices and resonators, or biological samples. In contrast s-SNOM imaging is more appropriate for imaging for solid-state physics studies and plasmonic setups, where spatial variations of fields and material properties on the sub-micrometer range are important. We have added a reference comparing the two techniques in detail [20].

It is for photoconductive antenna near-field setups that our technique offers the most immediate benefit, by enabling the resolution to be limited by noise rather than by how close the antenna can be brought to the sample.

However, more universally and beyond the practical benefit of our technique for photoconductive antenna setups, our method enables imaging with high resolution away from the sample, and so enables imaging in a way that does not perturb the field to be imaged itself: When bringing in a tip (be it a s-SNOM tip, a photoconductive antenna, or any other imaging device) into the near field of a sample, the fields are perturbed by the induced polarization of the tip [23,27-29,52]. While for many samples this may be a negligible effect, there are cases where this could destroy the very effect that is being measured. When weak, this perturbation is approximately linear and can be compensated for [23], but for strongly confined fields and highly resonant structures this is not the case: For example, when imaging plasmonic field confinement, a metallic SNOM tip could create a localized hot-spot severely distorting the local field distribution. Imaging the modal fields of a high-Q cavity with a tip in its near-field will lead to a scattering loss pathway that will reduce the quality factor and shift the frequency of the resonance [52]. This frequency shift scales linearly with the polarizability of the imaging tip (or roughly as the ratio of volume of the tip to volume of the modal field) – which for photoconductive tips sitting on a relatively large dielectric substrate can be sizeable – and scales quadratically with the field strength. Moving the imaging tip further away from the sample thus reduces the effect dramatically.

To highlight these benefits, we re-wrote most of the introduction, and parts of the abstract and conclusion – see attached bluelined manuscript, pages 3-5.

R1.0d This manuscript would be much more powerful if the authors devoted some time justifying the need for their approach, perhaps by focusing on these sorts of situation.

We thank the reviewer for this constructive suggestion: We have added in the Supplementary Material a study showing that the shift in frequency of a resonator due to the perturbation by an imaging tip can be reduced by orders of magnitude using our virtual superlens technique. This is not restricted to specific types of antennae, but universally applicable to any near field technique. (Supplementary Text, Section "Non-perturbative imaging" and Supplementary Figures 6-7).

R1.0e. Or, alternatively, some time speculating about how much farther their technique could be pushed (in terms of the trade-off posed here: spatial resolution vs. the distance between the sample and the measurement point).

To highlight the trade-off between distance, noise and resolution we have moved the section quantifying these effects from the Supplementary Material to the main text (Fig. 3 in the revised version of the manuscript).

R1.1 I am confused about the following sentence: "Figure 3(c) shows the average measured intensity across $y = 0 \pm 100 \mu\text{m}$ as a function of x and frequency, and highlights that apertures cannot be discerned **directly**, either due to diffraction at higher frequencies, or because of evanescent decay at lower frequencies." To me, it looks as if the two apertures are indeed well resolved at higher frequencies (although obviously not at lower frequencies). For example, it seems pretty clear when looking at Fig. 3(c) at 1.5 THz that there are two apertures, despite the weaker diffraction bump in between them. Not as clearly as in Fig. 3(f), but even so. The authors should clarify what they mean here.

That is a valid point that is worth clarifying, now addressed with the following sentence:

Figure 3(c) shows the average measured intensity across $y = 0 \pm 100 \mu\text{m}$ as a function of x and frequency, and highlights that apertures cannot be discerned directly: At higher frequencies, what would be measured by a scanning antenna or tip is a diffraction pattern that includes additional features, requiring phase reversal (i.e., lensing) to reconstruct an accurate image. At lower frequencies the evanescent decay blurs out the features of the double aperture.

R1.2 On page 4, top paragraph, second line from the end, where it says this: "i.e., setting regions where $k > k_0$ to zero" Is that supposed to be " $k_x > k_0$ "? Either that's a typo (with the subscript x accidentally omitted on the left hand side of the inequality) or I am confused by that sentence.

This was indeed a typographical mistake, we thank the reviewer for spotting it.

R1.3 I would mildly take issue with this phrase that appears in the very first paragraph of the document: "or nonlinear effects [8, 9], only really possible in the optical spectrum." Several groups have demonstrated nonlinear effects in near-field imaging and spectroscopy at terahertz frequencies. Indeed, there have even been suggestions that such nonlinearities can lead to improved spatial resolution compared to linear measurements. See, e.g., this reference: <https://doi.org/10.1364/OE.382130>

The Reviewer makes an interesting point. We were in fact referring to nonlinear effects allowing to beat the diffraction in the *far* field, such as through stimulated-emission-depletion fluorescence microscopy, not the use of nonlinear effects combined with a scanning tip in the *near* field. To avoid any misunderstanding, we reworded the sentence to clarify, and have added references [24,25] to the use of nonlinear effects in THz s-SNOM imaging:

Recent developments in the combined use of non-linearity and local optical excitation near s-SNOM tips could provide even further improvement in resolution [24,25].

[24] A. Pizzuto, D. M. Mittleman, and P. Klarskov, Laser THz emission nanoscopy and THz nanoscopy, *Optics Express* 28, 18778 (2020).

[25] A. Pizzuto, P. Ma, and D. M. Mittleman, Near-field terahertz nonlinear optics with blue light, *Light: Science & Applications* 12, 10.1038/s41377-023-01137-y (2023)

Reviewer #2

R2.0 I believe that this is a very useful approach, and can be readily used even in near field scanning systems for enhancing the resolution (that is also limited due to the finite distance from the probe to the sample).

We appreciate the reviewer's positive comments.

Reviewer #3

R3.1. What are the advantages of this work compared with terahertz near-field scanning imaging employing the AFM probes?

Reviewer 3 agrees with Reviewer 1 that our paper would benefit from clarifying this point– we have addressed this in responses to Reviewer 1, see our replies to queries R1.0a-e.

R3.2a. The proposed challenge “the resolution of the highest spatial frequencies is adversely affected by even modest losses” in the introduction seems to be shifted or unsolved in this work. The resolution in this work seems to be restricted by the noise, which is caused by the post-process method. Only low-loss conditions can be resolved.

This comment provides an excellent opportunity for clarifying one important advantage of our technique. The quoted sentence refers to *material* losses inevitable in conventional superlenses. Our approach circumvents material losses altogether by removing the need for any materials: the evanescent fields are measured in air rather than in a structured material designed to reverse evanescent decay, and the reversal of decay is achieved numerically instead. It is true that instead of losses, we are limited by the noise of the measurements, but this is much easier to reduce than material losses in material-based superlenses. Note the noise is not introduced by the method, it takes its origin in the measurement itself, but is then indeed amplified in post-processing. However, reducing measurement noise for example by longer averaging also leads to lower noise post processing and thus higher resolution - while material losses cannot be removed from material based-superlenses. We have added the following clarification in the Discussion:

Compared to previous superlens incarnations [34, 35, 42], our approach circumvents losses altogether by removing the need for materials: the evanescent fields are measured in air rather than after a structured material, and the reversal of decay is achieved numerically instead.

R3.3. To filter noise, the authors apply a low pass filter in k space. In this condition, the resolution is confined, since the high-k information is also lost. (for $k > 2.5k_0$). How to solve this problem?

This is indeed an intrinsic limitation of the method – to improve clarity, we have moved this discussion from the Supplementary Material to the main manuscript (see Fig. 3 of the revised manuscript and surrounding text). The highest spatial frequency is limited by a trade-off between antenna distance and signal-to-noise ratio. We used $k > 2.5k_0$ as an illustrative first example, but more generally general Eq. 5 gives the maximum spatial frequency resolvable for any distance and any level of noise. In our experimental example, the cutoff went up to $3.5k_0$ (see Fig. 5). In practice the signal to noise can, for example, be improved with longer averaging.

R3.3a. How can the phase be accurately measured in the terahertz region?

We use THz time-domain Fourier spectroscopy, which measures the time-dependent electric *field* (not intensity) directly. This gives the full phase information. We have added a section on the precision of the phase measurement as a function of frequency in Supplementary Material, and additional text in the Methods section of the main manuscript:

Fourier transforms of the temporal response at each pixel position provide the spectral information, including amplitude and phase (spectral resolution: 5-8 GHz). The phase fluctuation, obtained from the standard deviation of the phase of the Fourier transform of 850 consecutive measurements, is between 0.05-0.35 rad between 0.2-1.5 THz, as shown in Supplementary Fig. 5(a)-(c), which has no discernible effect on the superlens imaging quality for our experimental conditions, as shown in Supplementary Fig. 5(e)-(f).

R3.3b. How does the phase deviation affect your imaging?

An interesting question! To address this, and following our response to R3.3a, we have added Supplementary Fig. 5 to show that the method is surprisingly insensitive to phase noise – with field reconstruction remaining remarkably good even with phase noise well above the experimental phase noise level.

Supplementary Fig. 5. Phase fluctuation effects on terahertz imaging. (a) Example consecutive measurements of the terahertz electric field (850 measurements taken over 70 minutes) (b) Associated intensity for all 850 measurements, obtained from a Fourier Transform of each pulse in (a). (c) Phase fluctuation at each frequency, obtained from the standard deviation of the phase of each data point in (b). Associated effect of phase noise on superlensing for different levels of phase noise at (d) 0.18 THz, (e) 0.36 THz and (f) 0.67 THz, using the simulation data from Supplementary Fig. 4, adding phase noise as labelled, and with k_{\max}/k_0 as labelled. Within the measured range of phase noise, the effect of adding phase noise is small. The procedure becomes even more robust to noise for lower values of k_{\max}/k_0 , where the super lens procedure has smaller amplification factors increasing noise levels.

R3.4a The authors image the “THZ” words in their experiment, is it still effective to super-resolution image other samples?

While we use a single THz aperture for our experimental demonstration, we measure the *fields* transmitted through it over a wide range of frequencies. The field itself depends on the frequency, so each frequency in fact produces a different image: In Figure 5, E_x and E_y have different number of nodes and antinodes for each frequency. These match well with simulated fields at the aperture (Supplementary Figure 4) – so in effect we have demonstrated the imaging capability over 6 different field images. To clarify this point we have added the following text in the Experiments section:

As a result, for x-polarized fields the letters’ vertical (y-oriented) features are clearest, and clearly show different numbers of nodes and anti-nodes for different frequencies, revealing the frequency-dependent modal field structure inside each letter, in excellent agreement with simulations (see Supplementary Fig. 4). We repeat the above procedure for a polarization oriented in y relative to the sample orientation, and plot the corresponding $|E_y|^2$ in Fig. 5(c) to resolve the horizontal features of each letter.

The E_x and E_y components in Fig. 5(c,d) have the richest information on the actual, unperturbed local electric fields at the aperture site, but if the shape of the aperture rather than the fields is desired, a full image of the aperture can be obtained by summing the two contributions: Fig. 5(d) show the resulting $|E_x|^2 + |E_y|^2$ distribution, clearly showing the emergence of the letters “THZ” at all frequencies, down to $\lambda/7$.

R3.4b. The scattering spectrum by different samples will be different, so the low-pass filter may have different ranges.

We have not adapted the low-pass filter to the sample: The low pass filter cutoff is dependent on noise and distance/wavelength ratio (Eq. 5) which ultimately determines the resolution that can be achieved. For each field image at each frequency in Figure 5 different low-pass filters were used according to Eq. 5, indeed showing more detail when high resolution can be achieved. To clarify this we have moved the discussion on resolution/noise tradeoff into the main part of the manuscript, also in answer to Reviewer query R.3.3 and R.1.0.

R3.5 The realization of the spatial Fourier transforms in Figure 2,3 should be explained. The statement should include the step size of the space domain (Δx , Δy), the step size of the k domain, and the relationship between k_{max} and Δx , Δy .

We have made the data and code for obtaining Figures 2, 3, 4, 5, as well as Suppl. Fig. 5-6, publicly available. The private link to download the code is: <https://figshare.com/s/91a77322ddbfb08985ad>
This code explicitly presents the realization of both temporal and spatial Fourier transforms. In addition, we have added the following clarifying text and Supplementary Table 1 to the Supplementary Material.

SUPPLEMENTARY TABLE

Simulation and experimental spatial parameters

Supplementary Table I contains a summary of the step size in the spatial domain Δx and Δy , and the number of points N_x and N_y in x and y respectively, associated with the data shown in Figs. 2-5 of the main manuscript. The window size is given by $X_w = N_x \Delta x$ and $Y_w = N_y \Delta y$. The maximum spatial frequency for each coordinate can then be computed by $K_x^{\max} = \frac{\pi}{\Delta x}$ and $K_y^{\max} = \frac{\pi}{\Delta y}$, and the spatial frequency step size is given by $\Delta K_x = \frac{2\pi}{X_w}$ and $\Delta K_y = \frac{2\pi}{Y_w}$.

Data set	N_x	N_y	Δx [μm]	Δy [μm]
Double slit simulation (Figs. 2-3)	1000	N.A.	10	N.A.
Double aperture experiment (Fig. 4)	81	51	50	50
x -polarized "THZ" letters experiment (Fig. 5)	241	161	25	25
y -polarized "THZ" letters experiment (Fig. 5)	201	161	25	25
x -polarized "THZ" letters simulation (Suppl. Fig. 4)	601	401	10	10
y -polarized "THZ" letters simulation (Suppl. Fig. 4)	601	401	10	10

Supplementary Table I.

R.3.6. What is the difference between the reactive near-field region, radiating near-field region, and far-field region? Please explain their spatial division and analyze the characteristics of each area.

We adopted this terminology from the antenna literature, in particular from A. Yaghjian, "An overview of near-field antenna measurements", *IEEE Transactions on antennas and propagation* 34, 30 (1986) which we cite in the relevant paragraph. We have added a brief summary in the introduction as follows:

Tips of s-SNOMs oscillate within nanometers of the object, a distance well below $\lambda/2$, that is in the reactive near-field [26], where evanescent waves corresponding to high spatial frequencies haven't decayed much yet [26]. This is ideal to achieve high spatial resolution, but the proximity of the tips can also affect the fields to be measured. [...]

The radiating near-field is characterized by fields that are dominated by radiating rather than evanescent waves. In this region, the fields don't decay as $1/r$ yet as they do once in the far field, while high spatial frequencies decay significantly, preventing genuine sub-wavelength imaging [1].

Such increased measurement distances, while leading to a loss in resolution, have the benefit of reducing perturbations to the fields due to the probe itself, which can be important given NFPA's and their substrate are considerably larger than s-SNOM tips.

R.3.7. In the abstract, the authors say "At lower frequencies, exponentially decaying evanescent waves must be measured directly, requiring a tip or antenna to be brought into very close vicinity to the object." At terahertz frequencies, the evanescent waves can also be detected in the far field, without a tip or antenna. These conditions should not be excluded.

If the reviewer referring to s-SNOM type of techniques where the near field is scattered to the far field (by a tip in the near field), we hope that our new extended introduction and discussion of s-SNOM and other techniques clarifies the reviewer's concern. We take the opportunity to note that evanescent waves by definition decay exponentially – even in the closest "far" field at only 2 wavelengths away from the

sample, a wave with $k_x = 2.5k_0$ will have intensity reduced by $e^{-4\lambda\sqrt{(2.5k_0)^2 - k_0^2}} \simeq e^{-18.3\pi} \simeq 10^{-25}$. To detect, on average, a single THz photon per second at that distance would require the power to be 1kW at the source, and would need to scale exponentially with distances beyond 2λ . Therefore, detecting evanescent waves beyond the near field does require conversion to radiating waves with $1/r$ rather than evanescent dependence first, as done with s-SNOM techniques (or other techniques, e.g. nonlinear photoconversion).

REVIEWERS' COMMENTS

Reviewer #1 (Remarks to the Author):

The authors have adequately addressed all of my concerns.